# Apoptosis Deregulation and the Development of Cancer Multi-Drug Resistance

**DOI:** 10.3390/cancers13174363

**Published:** 2021-08-28

**Authors:** Christiana M. Neophytou, Ioannis P. Trougakos, Nuray Erin, Panagiotis Papageorgis

**Affiliations:** 1European University Research Center, Nicosia 2404, Cyprus; c.neophytou@research.euc.ac.cy; 2Tumor Microenvironment, Metastasis and Experimental Therapeutics Laboratory, Basic and Translational Cancer Research Center, Department of Life Sciences, European University Cyprus, Nicosia 2404, Cyprus; 3Department of Life Sciences, European University Cyprus, Nicosia 2404, Cyprus; itrougakos@biol.uoa.gr; 4Department of Cell Biology and Biophysics, Faculty of Biology, National and Kapodistrian University of Athens, 15784 Athens, Greece; 5Department of Medical Pharmacology, Cancer immunology and Immunotherapy Unit, Medical School, Akdeniz University, Antalya 07058, Turkey; nerin@akdeniz.edu.tr

**Keywords:** apoptosis, Bcl-2 family of proteins, cancer associated fibroblasts, cancer therapy, caspase-dependent death, epigenetic modifications, multi-drug resistance, PI_3_K/AKT pathway, tumor-microenvironment

## Abstract

**Simple Summary:**

Despite recent therapeutic advances against cancer, many patients do not respond well or respond poorly, to treatment and develop resistance to more than one anti-cancer drug, a term called multi-drug resistance (MDR). One of the main factors that contribute to MDR is the deregulation of apoptosis or programmed cell death. Herein, we describe the major apoptotic pathways and discuss how pro-apoptotic and anti-apoptotic proteins are modified in cancer cells to convey drug resistance. We also focus on our current understanding related to the interactions between survival and cell death pathways, as well as on mechanisms underlying the balance shift towards cancer cell growth and drug resistance. Moreover, we highlight the role of the tumor microenvironment components in blocking apoptosis in MDR tumors, and we discuss the significance and potential exploitation of epigenetic modifications for cancer treatment. Finally, we summarize the current and future therapeutic approaches for overcoming MDR.

**Abstract:**

The ability of tumor cells to evade apoptosis is established as one of the hallmarks of cancer. The deregulation of apoptotic pathways conveys a survival advantage enabling cancer cells to develop multi-drug resistance (MDR), a complex tumor phenotype referring to concurrent resistance toward agents with different function and/or structure. Proteins implicated in the intrinsic pathway of apoptosis, including the Bcl-2 superfamily and Inhibitors of Apoptosis (IAP) family members, as well as their regulator, tumor suppressor p53, have been implicated in the development of MDR in many cancer types. The PI_3_K/AKT pathway is pivotal in promoting survival and proliferation and is often overactive in MDR tumors. In addition, the tumor microenvironment, particularly factors secreted by cancer-associated fibroblasts, can inhibit apoptosis in cancer cells and reduce the effectiveness of different anti-cancer drugs. In this review, we describe the main alterations that occur in apoptosis-and related pathways to promote MDR. We also summarize the main therapeutic approaches against resistant tumors, including agents targeting Bcl-2 family members, small molecule inhibitors against IAPs or AKT and agents of natural origin that may be used as monotherapy or in combination with conventional therapeutics. Finally, we highlight the potential of therapeutic exploitation of epigenetic modifications to reverse the MDR phenotype.

## 1. Introduction

Novel diagnostic and cancer therapeutic technologies have improved patient response to treatment and have lowered mortality rates. However, in several cases, the 5-year survival rate remains low, mostly due to the intrinsic resistance or to the development of acquired resistance to anticancer drugs. Multi-drug resistance (MDR) refers to the state in which cancer cells become resistant to two or more drugs that have entirely different mechanisms of action and/or chemical structures. Thus, different cancer types become difficult to treat because of MDR. These mainly include breast, lung, colorectal and prostate cancer that represent the most frequently occurring malignancies with the highest mortality rates [1,2,3,4,5].

A variety of factors and mechanisms promote the development of MDR in cancer cells, including drug inactivation, detoxification mechanisms, increased drug efflux, mutations in genes encoding drug targets, epigenetic changes, deregulation of DNA damage/repair processes, contribution of cancer stem cells, increased tumor heterogeneity, involvement of the tumor microenvironment (TME), epithelial to mesenchymal transition (EMT), modulation of reactive oxygen species (ROS) and inhibition of cell death pathways [6]. Evasion of apoptosis or programmed cell death (PCD) has been well established as a one of the major hallmarks of cancer [7]. Deregulation of apoptotic pathways can lead to tumorigenesis, autoimmune and degenerative diseases [8]. In this review, we focus on the deregulation of apoptotic pathways and the development of multi-drug resistance in a variety of tumor types. We also discuss the role of the TME in regulating apoptosis in MDR tumors, as well as current and future therapeutic approaches targeting apoptosis that are being developed to overcome cancer MDR. Finally, we discuss the potential exploitation of epigenetic modifications for new therapeutic advances.

## 2. Overview of Apoptotic Pathways

Apoptosis is a tightly controlled physiological process, necessary for normal embryonic development, preservation of genome integrity, proper function of the immune system and maintenance of tissue homeostasis [9]. Apoptosis may be induced by a variety of agents, including low doses of radiation, hypoxia, heat, cytotoxic drugs or more specialized anti-cancer molecules. The apoptotic process does not induce inflammation as the cell contents are eventually absorbed by phagocytic cells [10,11].

Two main pathways contribute to apoptosis, i.e., (a) the intrinsic or mitochondrial pathway that is mostly activated by intracellular stress signals, including oxidative stress and (b) the extrinsic or death receptor pathway that is engaged following extracellular signals. The latter is induced following binding of death ligands to the extracellular domain of death receptors which (among others) include Receptor 1/Tumor Necrosis Factor-α (TNFR1/TNF-α) and Fas Receptor/Fas Ligand (FasR/FasL) [12,13,14,15,16]. Downstream to this binding, death receptors establish homotrimer structures followed by self-assembly of their intracellular parts that contain death domains (DDs) [17,18]. The intracellular domains of the TNF or FAS receptors then recruit adaptor proteins, including the TNFR1-associated death domain protein (TRADD) and FAS-associated death domain protein (FADD), respectively [19,20]; adaptor proteins contain the DED protein interaction component, that by recruiting inactive initiator caspase-8 forms the Death Inducing Signaling Complex (DISC) [21] which then triggers caspase-8 activation via oligomerization [22].

Caspases are cysteinyl, aspartate-specific proteases that play critical roles in apoptosis [23]. They are expressed as inactive pro-enzymes and contain an N-terminal pro-domain and a C-terminal catalytic domain. Their C-terminal domain comprises a p20 large subunit and a p10 small subunit [24]. The so-called initiator caspases (casp-2, -8, -9, -10) are activated in the early stages of apoptosis and induce a cascade of reactions to kickstart the apoptotic process. Executioner caspases (casp-3, -6, -7) are activated during the later stages of the process and are responsible for cleaving cellular components [25]. Structurally, the pro-domain of executioner caspases is very short compared to initiator caspases. Initiator caspases contain in their pro-domain either a death effector domain (DED) (caspases-8 and -10) or a caspase-recruitment domain (CARD) (e.g., caspases-2 and -9). DED is responsible for the interaction of caspases with molecules that regulate their activity [24]. Inactive caspases are activated via cleavage at aspartate residues, while the N-terminal domain is removed by cleavage between the large and small subunits [26]. Initiator caspases then cleave downstream caspases triggering a proteolytic cascade that amplifies the apoptotic signaling pathway.

The intrinsic apoptotic pathway is initiated by different signals, including ultra-violet (UV) or gamma irradiation, hypoxia, growth factors, hormone/cytokine deprivation, viral virulence factors, heat, DNA-damaging agents and the activation of oncogenic factors (Figure 1). Apoptotic signals induce a process called mitochondrial outer membrane permeabilization (MOMP) which involves opening of the mitochondrial permeability transition (MPT) pore and release of pro-apoptotic proteins from the inter-membrane space (IMS) into the cytosol [27]. Once the outer mitochondrial membrane becomes permeable, pro-apoptotic factors located in the inter-membrane mitochondrial space exit towards the cytosol. For example, cytochrome c and Apoptotic protease activating factor 1 (Apaf-1) are released, interact with caspase-9 and activate the latter, while forming a structure known as the apoptosome [28,29]. Further, Smac/DIABLO and HtrA2/Omi are released and induce apoptosis by inhibiting a group of proteins called Inhibitors of Apoptosis Proteins (IAPs) [30,31].

MOMP is controlled by Bcl-2 family members. Bcl-2 proteins are categorized into three different groups according to their function and number of BH domains present in their structure: 1. anti-apoptotic members including Bcl-2, Bcl-xL and Mcl-1 that contain three or four BH domains, 2. pro-apoptotic members such as Bax and Bak that contain BH1, BH2 and BH3 and 3. pro-apoptotic BH3-only members including Bad, Bid, Noxa, Puma and BNIP3 [32,33]. The anti-apoptotic members can bind to BH3-only members through hydrophobic grooves formed by their BH domains [34,35]. This interaction determines the activation status of this class of proteins. BH3-only members induce apoptosis by blocking the function of anti-apoptotic family members and/or by interacting with and activating pro-apoptotic proteins such as Bax and Bak [36]. Anti-apoptotic Bcl-2 family proteins, block apoptosis by inhibiting the activity of pro-apoptotic proteins and preventing MOMP [34].

During apoptosis, anti-apoptotic protein levels decrease, while the levels of pro-apoptotic members rise; in fact, a decrease in the Bcl-2/Bax ratio is considered a reliable indicator of apoptosis. Furthermore, since the Bcl-2 family regulates mitochondrial permeability, their subcellular localization changes during apoptosis. For example, following apoptotic stimuli, Bax translocates from the cytosol to mitochondria [34]. The extrinsic pathway can also induce mitochondrial apoptotic pathways since caspase-8 can cleave Bid (a Bcl-2 pro-apoptotic protein) to its active form namely tBid which promotes MOMP [37].

Caspases -3, -6 and -7 are considered “effector” caspases and cleave cellular products during the later stages of apoptosis [38]. Caspase-9 and -8 can cleave caspase-3, while caspase-7 is a downstream target of caspase-9. Caspase-3 can also activate caspase-6 [39]. Active effector caspases break down many substrates ultimately causing DNA cleavage as well as nuclear and cytoskeletal protein degradation. The endonuclease Caspase-Activated DNase (CAD) is physiologically bound to ICAD (Inhibitor of Caspase-Activated DNase), a substrate of caspase-3. Upon its activation, CAD fragments DNA at ~180-bp pieces [40]. PARP-1 is another caspase substate; normally, it participates in DNA repair mechanisms, but it is also involved in DNA replication and transcription, cellular repair, cytoskeletal organization and protein degradation [41]. PARP-1 breakdown by caspases is crucial during apoptosis. The degradation of these targets ultimately leads to the biochemical and morphological changes observed in apoptotic cells including cell shrinkage, cytoplasmic condensation and generation of apoptotic bodies [42,43]. At the final phase of apoptosis, phagocytic cell receptors recognize ligands presented on the surface of apoptotic bodies causing their destruction by professional phagocytic cells [9]. The deregulation of apoptotic pathways that ultimately leads to MDR is described below.

## 3. Deregulation of the Intrinsic Apoptotic Pathway in MDR Tumors

A particularly important mechanism which promotes cancer cell resistance to chemotherapy is inhibition of apoptosis [44]. Proteins involved in the intrinsic pathway of apoptosis, including Bcl-2 family members and the tumor-suppressor p53, are commonly deregulated in MDR cancers, whereas IAPs, which control caspase activation, are often overexpressed. In addition, related survival pathways, such as PI_3_K/AKT, often contribute to the development of resistance.

### 3.1. Bcl-2 Family Deregulation in MDR

Several proteins involved in the intrinsic pathway of apoptosis have been identified as important cellular oncogenes that not only promote tumorigenesis but also contribute to anti-cancer drug resistance. Inactivating mutations or deletions of pro-apoptotic Bax or Bak are rare, but many cancers, especially those being refractory to therapy, such as colon, gastric and leukemia, overexpress one or more pro-survival family members, including Bcl-2, Bcl-xL and Mcl-1 [45,46,47,48,49]. Initial studies in Bcl-2 transgenic mice revealed accumulation of lymphocytes resistant to diverse cytotoxic agents, including chemotherapeutic drugs [50,51,52]. Multiple subsequent studies indicated that high levels of Bcl-2 gene expression correlate with severity of malignancy in cancer patients, including melanoma, breast, prostate, small cell lung, colorectal and bladder cancer, while increased Bcl-2 expression is associated with resistance to chemotherapy and radiation [53].

Another gene implicated in chemoresistance, is the tumor suppressor p53 which controls the transcription of numerous genes involved in DNA repair, metabolism, cell cycle arrest, apoptosis and senescence [54]. One of the first physiological p53 functions described was its ability to induce apoptosis in transformed cells [55]. p53 transcriptionally upregulates the expression of apoptotic-related proteins, such Puma, Noxa, Bid and Bax and can also physically interact with and neutralize the anti-apoptotic activity of Bcl-2 and Bcl-xL [56]. Thus, p53 has a dual role both as a sensitizer, as well as an activator of apoptosis. p53 mutations, mainly missense mutations, repress apoptosis thus causing therapeutic resistance [57]. Notably, mutant p53 can also inhibit apoptosis through the caspase-dependent apoptotic singling cascade [58]. Overexpression of mutant p53 has been correlated with resistance to conventional drugs including cisplatin, antimetabolites (gemcitabine), anthracyclines, (doxorubicin), alkylating agents (temozolomide) and drugs with specific targets such as EGFR-inhibitors (cetuximab) and antiestrogens (tamoxifen). In addition to mutations in *TP53* gene causing protein conformational changes, p53 activity may also be impaired due to alterations in p53-regulating proteins, such as MDM2 [59]. Therefore, several small molecules, that accelerate mutant p53 protein turnover or convert it into the wild-type conformation, have been developed and applied in clinical therapy [54]. However, targeting p53 in tumor cells often leads to several side effects and drug cytotoxicity in normal tissues [60]. Importantly, ROS regulate p53 activity by oxidizing the cysteine residues present in its structure. This modification leads to p53 inability to bind to DNA and activate specific genes [61]. 

### 3.2. Inhibitors of Apoptosis Proteins (IAPs) and Their Role in MDR

IAPs are a class of proteins that are frequently overexpressed in human cancers conveying resistance to apoptosis and therapy. IAP family members include Survivin, X-linked inhibitor of apoptosis (XIAP), inhibitors of apoptosis 1 and 2 (c-IAP1 and c-IAP2), BIR-repeat-containing ubiquitin-conjugating enzyme (BRUCE/Apollon), neuronal apoptosis inhibitor protein (NAIP), IAP-like protein 2 (ILP-2) and melanoma IAP (ML-IAP/Livin) [62]. IAPs contain one or more baculovirus inhibitor repeat (BIR) domains, an amino-terminal, 70-residue structure with distinct functions. In XIAP, the region between BIR1 and BIR2 specifically targets caspases -3 and -7 while BIR3 inhibits the activity of caspase-9 [63]. The RING domain, located in XIAP, Livin, ILP2, c-IAP-1 and c-IAP-2 protein structures, catalyzes the ubiquitination and proteasomal degradation of target proteins. c-IAP and c-IAP-2 are critical regulators of the noncanonical NFkB pathway and promote malignancy by inducing the degradation of NFkB-inducing kinase (NIK) [64].

IAPs were originally thought to physically bind and block caspase activity, inhibiting both the extrinsic and intrinsic pathway of apoptosis [65]. Some family members, including c-IAP1 and c-IAP2, have a caspase recruitment domain in their structure. However, with the notable exception of XIAP, they cannot directly bind and inhibit caspases [66,67]. Under physiological conditions, IAP activity is controlled by Smac/DIABLO and Omi/HtrA2 that are released by the mitochondria and diminish their caspase-inhibitory effects. Overexpression of IAPs has been reported in many human cancers and has been correlated with resistance to therapy and worsening disease [67,68]. High levels of c-IAP1, c-IAP2, XIAP, Survivin and NAIP have been reported in breast cancer [69]. Furthermore, during early stages of pancreatic cancer, elevated levels of c-IAP2 contribute to malignant progression [70]. In esophageal cancer, increased XIAP levels inhibit caspase-3 activation and lead to apoptosis resistance [71]. Targeting IAP family members with agents that act as “SMAC mimetics” is widely investigated as a promising anti-cancer approach against MDR cancers [72].

Survivin is the smallest IAP protein. It is physiologically expressed during embryonic development to inhibit apoptosis and promote proliferation in developing tissues [73,74]. Survivin is expressed at very low levels in differentiated tissues. However, it is overexpressed in most primary tumors and has been correlated with resistance to chemotherapy and radiotherapy-induced cell death as well as poor prognosis [75,76,77,78,79,80]. Increased Survivin expression in cancer cells is partially attributed to aberrant activation of upstream survival pathways, such as NFkB, which transcriptionally upregulate Survivin [81]. The anti-apoptotic mechanism of Survivin involves formation of a complex with XIAP, that protects XIAP from ubiquitin-dependent degradation and increases its caspase-inhibiting function [82]. Furthermore, Survivin may sequester Smac/DIABLO away from XIAP or inhibit Smac/DIABLO translocation from the mitochondria to prevent XIAP inactivation [83,84]. Therapeutic exploitation of Survivin is pivotal, as it represents a cancer cell-specific drug target. However, Survivin-targeting agents have performed poorly in clinical studies, highlighting the need for developing novel approaches against this protein [85].

### 3.3. PI_3_K/AKT Pathway in Multi-Drug Resistance

The synergy between apoptosis resistance and increased survival signaling is critically important in cancer development. The PI_3_K/AKT pathway responds to a variety of external signals and is involved in the regulation of different cellular functions, including cell cycle progression, survival, metabolism, gene transcription and maintenance of DNA integrity [86]. Deregulation of this pathway has been implicated in MDR of many cancers, including leukemia, hepatocellular carcinoma, breast cancer, ovarian cancer, lung cancer and melanoma [87,88,89,90,91,92]. 

Binding of growth factors to receptor tyrosine kinases (RTKs) stimulate PI_3_K by autophosphorylation which leads to the phosphorylation and activation of serine/threonine kinase AKT (Protein Kinase B, PKB). RTK-PI_3_K complexes localize at the cell membrane where the PI_3_K subunit, p110, catalyzes the conversion of Phosphatidylinositol 4,5-bisphosphate PtdIns(4,5)P_2_ (PIP_2_) to Phosphatidylinositol (3,4,5)-triphosphate PtdIns(3,4,5)P_3_ (PIP_3_) [93]. Subsequently, AKT travels to the plasma membrane where it becomes phosphorylated. The tumor suppressor phosphatase and tensin homology deleted on chromosome 10 (PTEN) indirectly hinders AKT activity by converting PIP3 to PIP2 [94]. The phosphorylation of PTEN preserves its stability [95]. Phosphoinositide-dependent kinase-1 and -2 (PDK1 and PDK2) are responsible for activating AKT via phosphorylation on residues Thr^308^ and Ser^473^, respectively [96,97,98]. Phosphorylation of Thr^308^ partially activates AKT, while phosphorylation of both sites is required for its full activation [99]. Once activated, p-AKT translocates to the cytosol or the nucleus where it phosphorylates and therefore modulates the function of downstream substrates, including several targets being implicated in cancer initiation and progression. AKT enhances cell survival by negatively regulating the function or expression of pro-apoptotic proteins that inactivate Bcl-2 family members. Additionally, AKT promotes survival by eliciting p53 degradation [100].

A large percentage of tumors carry alterations in PI_3_K, AKT or PTEN leading to MDR. The sustained expression of pro-survival signals renders cancer cells resistant to anticancer agents. Phosphorylated AKT enhances cell survival by phosphorylating many proteins implicated in apoptotic pathways, including glycogen synthase kinase-3 (GSK-3), forkhead transcription factors (FOXO), caspases and proteins implicated in NFkB signaling [101]. AKT activates anti-apoptotic members of the Bcl-2 family, such as Bcl-2 and Bcl-xL, and IAPs including Survivin and XIAP via phosphorylation of the transcription factor cyclic AMP response element-binding protein (CREB) and IkB kinase (IKK), a positive regulator of NFkB [102,103,104,105]. Increased activity of the PI_3_K/AKT pathway attenuates chemotherapy-induced apoptosis by diminishing the levels of pro-apoptotic Bax and increasing the levels of anti-apoptotic Bcl-2 and XIAP [106]. In addition, AKT phosphorylates the pro-apoptotic protein Bad at Ser^136^, thus hindering its interaction with Bcl-xL and allowing the anti-apoptotic function of the latter. GSK-3, in response to insulin, regulates glycogen synthesis which has been shown to regulate cyclin D1 proteolysis and subcellular localization. GSK-3 activity is inhibited by AKT-mediated phosphorylation Ser^21^ [107,108,109]. AKT also activates the MEK-MAP kinase pathway in cancer cells promoting survival and proliferation, by phosphorylating cRAF at multiple amino acid residues, controlling its activity. [110]. Importantly, AKT can directly inhibit the caspase cascade; pro-caspase-9 is an AKT substrate and can be phosphorylated on Ser^196^ [111]. Moreover, the phosphorylation of FOXO transcription factors by AKT causes their degradation in the cytoplasm and enhances cell survival by blocking the transcription of death receptor ligands TRAIL and Fas, and of pro-apoptotic Bcl-2 members Bim and BNIP3 [112]. The implication of FOXOs in the development of MDR is highlighted by their key role in regulating drug efflux pump ABCD1 in leukemic and breast cancers and by eliciting resistance to agents that act via the accumulation of ROS [113]. A deeper understanding of the role of FOXOs in these two processes will enable the development of effective therapeutics. 

Often, upstream and downstream proteins should also be deregulated to achieve MDR in cancer cells. Upstream regulators of PI_3_K, RTKs HER-2 and EGFR were found to be amplified in human cancers. HER-2 is overexpressed in 20–30% of primary breast cancers that also exhibit constitutive AKT activity, while EGFR overexpression has been reported in breast, lung and colorectal carcinoma and glioblastoma [114,115,116]. Notably, almost 70% of endometrial and ovarian cancers harbor activating mutations of PIK3CA, the gene that encodes the p110α catalytic subunit of PI_3_K [106]. PIK3CA mutation has been associated with elevated PI_3_K and AKT activity [117,118]. A mutated form of the PIK3CA protein was also found to selectively phosphorylate AKT and FOXO promoting cellular growth and cancer cell invasion [119]. The AKT2 gene is often amplified in human cancers, including lung and ovarian cancers while both AKT1 and AKT2 gene amplification has been reported in breast and colorectal cancers [120,121,122,123,124]. As previously mentioned, PTEN is responsible for the indirect inactivation of AKT by converting PIP3 to PIP2, thus acting as a tumor suppressor. Loss of PTEN can occur either via gene mutation, deletion or promoter hypermethylation leading to elevated concentrations of the PIP3 substrate. Consequently, downstream components of the PI_3_K pathway, including AKT and mTOR, are constitutively active [94,125]. Based on its implication in cancer cell progression, the PI_3_K/AKT pathway has been extensively studied as a promising drug target against malignant progression [126]. 

## 4. Implications of the TME in Apoptosis and MDR

The TME plays a crucial role in tumor growth, metastasis and development of MDR. The TME consists of immune cells, fibroblasts and endothelial cells that communicate with cancer cells through paracrine signaling [127]. Immune cells present in the TME can activate or inhibit apoptotic pathways and affect response to therapy [128]. Cytotoxic lymphocytes induce the activation of effector mechanisms, such as release of death ligands such as FasL and TRAIL [129,130], as well as activation of granule exocytosis pathway [131,132]. Neutrophils and monocytes express TRAIL and target TRAIL receptor-expressing tumor cells [133]. Macrophages, the major phagocytic cells of the innate immune system, can also induce TRAIL-mediated apoptosis of cancer cells [134]. Cytokines, such IFNs, CD137 and IL-24, secreted by tumor-associated immune cells promote apoptosis in cancer cells [133,135,136,137]. In addition, non-cellular TME constituents are important mediators of cancer cell behavior, such as excess extracellular matrix (ECM) deposition, which compresses blood vessels and reduces perfusion, as well as acidic and hypoxic milieu, which collectively impair drug delivery [138,139,140]. Here, we mainly focus on the cellular components of the TME that regulate apoptosis of cancer cells and, more specifically, cancer associated fibroblasts (CAFs) which alter the apoptotic responses of cancer cells to cytotoxic drugs (Figure 2).

### 4.1. Cancer Associated Fibroblasts in Apoptosis and MDR

During carcinogenesis, CAFs are activated as a major component of the tumor stroma [144,145]. CAFs secrete various extracellular matrix proteins, chemokines, cytokines, as well as growth factors and extensively contribute to tumor progression, invasion and metastasis [145,146]. CAFs are also linked to poor survival in most cancers and are considered potential therapeutic targets [147]. Factors released by CAFs increase tumor cell survival via the activation of anti-apoptotic pathways or by induction of the epithelial to mesenchymal transition (EMT) and cancer stem cell (CSC) phenotype, as demonstrated in melanoma, non-small cell lung cancer (NSCLC) and colorectal cancer [148,149,150,151,152,153].

#### 4.1.1. CAF-Derived Extracellular Vesicles

The significance of extracellular vesicles (EVs) derived from CAFs (CAF-EVs) in the progression of carcinomas and resistance to apoptosis has been increasingly recognized. CAFs secrete exosomes that are loaded with proteins, lipids and RNAs to functionally control the properties of the TME [154]. CAF-EVs exert tumor-promoting functions and microRNAs (miRNAs) found in CAF-EVs participate in the interaction between cancer cells and CAFs [154,155,156]. Apoptosis of breast cancer cells is inhibited by CAF-EVs that reduce miR-30e expression to upregulate collagen triple helix repeat containing 1 (CTHRC1); this, in turn, activates the Wnt/β-catenin pathway to facilitate breast cancer development and progression. Overexpression of miR-30e or silencing of CTHRC1 suppresses proliferation, migration/invasion of breast cancer cells and promotes apoptosis [154]. CAF-derived exosomes could also transfer miR-181d-5p to enhance breast cancer aggressiveness. CAFs antagonize apoptosis in MCF-7 cells via transfer of miR-181d-5p which downregulates homeobox A5 (HOXA5) and caudal-related homeobox 2 (CDX2) [157]. Long non-coding RNAs (lncRNAs) are a heterogeneous class of transcripts longer than 200 nucleotides with limited protein-coding potential [158,159]. CAFs were shown to transfer lncRNA H19 to neighboring colorectal cancer cells [151,160,161,162] and promote the stemness via activating Wnt/β-catenin signaling causing resistance to oxaliplatin-mediated apoptosis [163]. Exosomal lncRNA enhance invasion, migration, proliferation and inhibit apoptosis in cervical cancer cells and in NSCLC [164,165]. The importance of lncRNAs in regulating drug resistance in cancer cells has recently been described in a detailed review [166]. 

#### 4.1.2. Interleukins Secreted by CAFs

Lung adenocarcinoma is one of the most CAF-rich cancers. The role of CAFs in resistance to chemotherapy of lung adenocarcinoma is well appreciated [167,168]. Cisplatin treatment increases Interleukin-11 (IL-11) levels in CAFs which protects lung adenocarcinoma cells from apoptosis [169]. Cisplatin induces DNA damage and subsequently activation of apoptosis [170]. IL-11, a member of IL-6 family, binds to IL-11Ra2 to activate signaling. [171]. Patients with high IL-11Rα levels show poor response to cisplatin-based chemotherapy [169]. IL-11 induces Signal transducer and activator of transcription 3 (STAT3) phosphorylation and increases the expression of anti-apoptotic protein Bcl-2 and Survivin in cancer cells. As mentioned above, Bcl-2 and Survivin have been extensively implicated in the development of chemoresistance in cancer [172,173]. The anti-apoptotic effects of IL-11 can be prevented by suppressing STAT3 phosphorylation or silencing IL-11Rα expression in lung adenocarcinoma [169].

IL-6 is another key cytokine, secreted by cancer cells, immune cells and CAFs, which inhibits apoptosis of cancer cells through STAT3 activation [174,175,176]. IL-6 binds to the cell surface receptor glycoprotein 130 (gp130) and activates several cell survival-related pathways promoting chemotherapeutic resistance in breast, ovarian and endometrial cancers [177,178,179]. Significant amount of IL-6 in the TME originates from CAFs and is involved in carcinogenesis and metastasis [180,181]. IL-6 derived from CAFs prevents chemotherapy-induced apoptosis by increasing the phosphorylation of Jak1 and STAT3, and the expression of the anti-apoptotic proteins Bcl-2 and Survivin in gastric carcinoma [182]. Studies conducted in CAFs derived from human gastric carcinomas further demonstrated the role of CAFs in prevention of early apoptosis of gastric cancer cells treated with 5-fluorouracil (5-FU). Factors secreted from CAFs not only inhibit apoptosis but also induce an EMT phenotype in gastric carcinoma [183]. Treatment with conditioned medium from activated gastric carcinoma CAFs decreases response of gastric cancer cells to 5-FU by suppression of apoptosis-related proteins, such as Bak, Bax, cleaved caspase 3 and cleaved PARP [183]. Similarly, chemotherapy-induced apoptosis of pancreatic cancer cells significantly decreases in the presence of CAFs [184,185]. Overall, the role of IL-6, IL-11 as well as of other interleukins, has been observed in MDR cancer cells; the clinical attempts to block their effects for therapeutic intervention have been described in a recent review [186]. 

#### 4.1.3. Regulation of Sex Determining Region Y-box 2 by CAFs

Sex determining region Y -box 2 (Sox2), an essential embryonal stem cell transcription factor, may also play an important role in CAF-induced drug resistance. Sox2 is linked to the formation and maintenance of CSC phenotype and is implicated in drug resistance and poor patient prognosis [187,188,189,190,191,192]. In ER^+^ breast cancer cells, a mixture of CAF-secreted factors strongly induced Sox2 expression. In addition, Sox2 blocked apoptosis, enabled cellular growth and shielded cells against the anti-estrogen fulvestrant [193]. CAFs also minimized the effectiveness of tamoxifen in breast cancer cells [194]. Whereas Sox2 is an attractive therapeutic target, direct targeting of Sox2 via siRNA has shown poor outcomes due to inefficient delivery and efficacy; novel approaches include the design of artificial transcription factors (ATFs), that bind to proximal *SOX2* promoters and reduce its expression [195].

#### 4.1.4. Growth Promoting Proteins Released by CAFs

CAFs secrete hepatocyte growth factor (HGF) that mediates resistance to cancer cell apoptosis [196]. HGF also induces cell proliferation, cancer cell motility and migration. Many cancer types, including ovarian, gastric, colorectal and pancreatic, overexpress c-Met, which functions as a specific HGF receptor [197,198]. HGF-induced c-Met activation triggers downstream the PI_3_K/Akt pathway, enabling cancer progression [199,200,201]. In lung cancer, HGF derived from CAFs, attenuates the apoptotic effects of paclitaxel (PAC) by upregulating glucose-regulated protein 78 (GRP78) [167]. GRP78 acts as a chaperon protein in the endoplasmic reticulum (ER) where it regulates protein folding; it is highly expressed on the surface of cancer cells [202] and enables malignant growth, motility, migration and resistance to therapy [203,204]. In ovarian cancer, HGF secreted by CAFs attenuated paclitaxel-induced apoptosis by activating the c-Met/PI_3_K/Akt pathway and signaling involving GRP78 [196]. In addition, GRP78 inhibits apoptosis by interacting with caspase-7 or p53 [205,206] and also by binding to Bax and Bik to prevent mitochondrial release of cyt-c [207,208]. This evidence suggests that the activation of PI_3_K/Akt and GRP78 may be implicated in cancer progression and resistance to therapy. Activation of PI_3_K/Akt pathway by other CAF-originated factors can also lead to resistance to apoptosis induced by cytotoxic drugs. More specifically, the CAF-derived chemokine CCL5, promoted cisplatin resistance in ovarian cancer cells by affecting the PI_3_K/Akt signaling pathway [209,210]. Netrin-1 is a multifunctional secreted glycoprotein upregulated in various cancers, such as gastric and lung, and may inhibit apoptosis induced by the dependence receptors DCC and UNC5H [211,212]. Netrin-1 and its receptor (UNC5B) are upregulated in CAFs of lung and colon tumors [213]. Inhibition of netrin-1 abrogates CAF-mediated increase in cancer stemness [213]. Hence, Netrin-1 secreted by CAFs may play an important role in inhibition of apoptosis and drug resistance. 

TP53-regulated inhibitor of apoptosis 1 (TRIAP1) is a small, 76-amino acid long, evolutionary conserved protein which inhibits apoptosis and promotes DNA repair [214,215]. Loss of caveolin-1 in CAFs augmented the secretion of TRIAP1 from CAFs causing radiation resistance of prostate cancer cells by hindering apoptosis [216]. Similarly, knockdown of TRIAP1, using microRNA miR-320b, induced mitochondrial apoptosis [214,217].

Folicular lymphoma-associated CAFs, isolated from malignant lymphoma patients, were shown to protect tumor cells from apoptosis in response to cytotoxic drugs [218]. These CAFs do not alter proliferation rate of cancer cells but markedly upregulate the expression of the anti-apoptotic BCL2L1 gene in folicular lymphoma cells [219,220,221]. Midkine can also mediate CAF-induced inhibition of apoptosis and chemoresistance. Midkine is a heparin-binding growth factor and induces tumor progression by enhancing carcinoma cell growth, survival [222,223], invasiveness, migration, and chemotherapy resistance [224,225]. Primary CAFs from oral squamous cell carcinoma secrete high levels of midkine, which abrogate cisplatin-induced cell death [226]. Midkine also enables glioma cells to become resistant to tetrahydrocannabinol by obstructing the ALK receptor and inhibiting autophagy-mediated cell death via the Akt/mTORC1 pathway [227]. Midkine induces the expression of lncRNA ANRIL in cancer cells while lncRNA ANRIL knockdown blocked proliferation and promoted apoptosis to augment cisplatin cytotoxicity via impairment of the drug transporters MRP1 and ABCC2 [226]. Furthermore, knockdown of lncRNA ANRIL increased the activation of caspase-3 and inhibited Bcl-2 expression [226].

Conclusively, CAFs not only enhance the aggressiveness of cancer cells but also render them resistant to therapy-induced apoptotic effects by secreting various factors including miRNAs, lncRNAs, cytokines and chemokines. All these CAF-secreted factors induce activation or upregulation of factors implicated in apoptosis inhibition and overactivation of survival pathways in cancer cells in response to chemotherapy. Given the crucial roles of CAFs in carcinogenesis and drug resistance, better understanding of the underlying mechanisms will uncover novel targets to overcome drug resistance mediated via deregulation of cell death pathways.

## 5. Therapeutic Approaches to Induce Apoptosis in MDR Cancers

Potent apoptosis-inducing approaches can prevent tumor initiation and progression. Many proteins involved in apoptosis have been targeted with small molecule inhibitors, epigenetic drugs and natural or synthetic compounds (Table 1). These agents may be used as monotherapy, but they have been often evaluated in combination with other targeted or conventional anti-tumor therapeutics. The Bcl-2 family of proteins represent an attractive target for therapy as it is often deregulated and confers resistance in cancer. Consequently, small molecule inhibitors that can interact with BH3 domains and antisense oligonucleotides have been developed [228,229]. These small molecules against Bcl-2 proteins can be categorized as BH3 mimetics (i.e., ABT-737, ABT-263) and small molecules with BH3 putative mimetic action (i.e., gossypol, obatoclax etc.) [230].

**Table 1 cancers-13-04363-t001:** Types of anti-cancer treatments against multi-drug resistance involving apoptotic pathways.

Therapeutic Class	Compound	Observed Effect	Model	Ref
Small molecule inhibitors	Venetoclax (Bcl-2 inhibitor)	Directly blocked the wild-type ABCG2 efflux function and inhibited the ATPase activity of ABCG2.	Human embryonic kidney cell line HEK293 overexpressing ABCG2 in vitro.	[231]
	ABT-737 (BH3-mimetic)	In combination with Fenretinide, synergistically induced cyt-c release, activation of caspases, Bax, t-Bid and Bak.	MDR neuroblastoma cell lines in vitro.	[232]
	Nutlin5 (MDM2-p53 antagonist)	Reversed MDR-1-mediated multidrug resistance in a p53-independent manner.	High MDR-1-expressing p53 mutant neuroblastoma cell lines in vitro.	[233]
	MI-219 (MDM2 inhibitor)	Sensitized cells to androgen ablation and radiotherapy by inducing DNA damage and apoptosis.	Prostate Cancer Cells in vitro.	[234]
	Thiosemicarbazone	Inhibited cell cycle progression at the G1 phase.	MCF7 and MCF7/ADR cells in vitro.	[235]
	LY294002 (PI3K inhibitor)	Inhibited the expression of p-Akt and P-gp.	Leukemia cell line K562/DNR and gastric cancer cell line SGC7901/ADR in vitro.	[236]
	Metformin (Metabolic inhibitor)	In combination with 2-deoxyglucose selectively enhanced cytotoxicity of Doxorubicin leading to G2/M arrest and apoptosis.	MCF-7/Dox breast cancer cells in vitro.	[237]
	BEZ235 (PI3K/mTOR inhibitor)	Caused a dose-dependent decrease in cell viability in combination with Dox, associated with an increase in cleaved PARP.	Ovarian A2780 and pancreatic MiaPaca2 cancer cells in vitro.	[238]
	AZ D8055 (mTORC1/2 inhibitor)	Inhibition of mTOR and caspase-3 cleavage in platinum-resistant cells.	Advanced-stage ovarian clear cell carcinoma patient-derived xenograft models.	[239]
	Rapamycine (mTOR inhibitor)	Inhibited PI_3_K/AKT pathway, blocked proliferation, sensitized cells to Tamoxifen and Fulvestrant.	Breast cancer cells resistant to endocrine therapy in vitro.	[240]
	YM155 (Survivin inhibitor)	Survivin depletion and p53 activation.	Neuroblastoma cell lines and their sublines with acquired resistance to clinically relevant drugs in vitro.	[241]
Natural agents and derivatives	Wagonin	Promoted TRAIL-induced apoptosis in vitro and downregulated the expression of anti-apoptotic XIAP, cFLIP_L_, cIAP-1 and cIAP-2.	Non-small cell lung cancer in vivo.	[242]
	Luteolin	Generated ROS leading to DNA damage and activated the ATR/Chk2/p53 pathway independently of the P-gp efflux pump.	MDR breast cancer cells in vitro.	[243]
	Fisetin	Concurrent treatment with chemotherapeutic drugs activated caspases -8 and -3, release of cyt-c and inhibited survival pathways IGF1R and AKT.	Colon cancer cells resistant to both Irinotecan and Oxaliplatin in vitro/in vivo.	[244]
	Genistein	Pre-treatment inhibited NFkB activity and led to increased growth inhibition and apoptosis in combination with Cisplatin and Docetaxel.	Prostate and lung cancer cells in vitro/in vivo.	[245]
	Resveratrol	- Induced apoptosis by upregulating miR-34c and p53.	- Platinum-resistant colorectal cancer cells, in vivo.	[246]
		- Reversed MDR by targeting Survivin and activating caspase-3.	- Non-small cell lung MDR cancer cells, in vivo.	[247]
	Curcumin	- Sensitized cells to capecitabine by inhibiting NFkB, reduced Bcl-2, IAP-1, Survivin, COX-2, MMP-2, ICAM-1, CXCR4 and VEGF	- Colorectal cancer to capecitabine in vivo	[248]
		- Difluorinated Curcumin downregulated PTEN inhibitor, miR-21.	- Colorectal cancer cells resistant to 5-FU and oxaliplatin in vitro.	[249]
		- In combination with EGCG led to synergistic effects through activation of the caspase-dependent signaling pathway, and downregulation of Bcl-2 and Survivin.	- Resistant breast cancer cells in vitro.	[250]
	Ellagic acid	In combination with 5-FU increased the Bax/Bcl-2 ratio, caused changes in mitochondrial membrane potential, activated caspase-3 and induced apoptosis.	Colorectal cancer cells in vitro.	[251]
	O-methylated coumarin	Inhibited the PI_3_K/Akt signaling pathway.	Myelogenous leukemia K562/ADM cells in vitro.	[252]
	Vitamin E and derivatives	TPGS induced cell cycle arrest and apoptosis selectively in Survivin-overexpressing breast cancer cells.	Breast cancer cells in vitro.	[253]
TME/Immune regulation	Pirfenidone	Induced apoptosis in CAFs at high concentration; at low concentrations induced apoptosis and decreased tumor progression synergistically with Cisplatin.	NSCLC cells in vitro and in vivo.	[254]
	Combination of anti–CTL-4 plus anti–PD1 therapy	Mediated a switch from expansion of phenotypically exhausted CD8^+^ T cells to expansion of activated effector CD8^+^ T cells.	Melanoma patients.	[255]
	Combination of EGFR-TKIs and anti-PD-1/PD-L1 antibodies	PD-L1 mediated by EGFR activation could induce the apoptosis of T cells through PD-L1/PD-1 axis in tumor cells.	EGFR-TKIs-resistant NSCLC cells with EGFR mutation in vitro.	[256]
	MEDI9447	Antibody targeting ectoenzyme CD73, increased CD8^+^ effector cells and activated macrophages.	Mouse syngeneic colorectal tumor growth in vivo.	[257]
Epigenetic drugs	Hydralazine (DNMTi)	In combination with Magnesium Valproate LP improved progression-free survival.	Metastatic Recurrent or Persistent Cervical Cancer patients.	[258]
	Parthenolide (HDACi)	NFkB and HIF1-α Inhibition.	Brain, breast, colon cancer cell lines in vitro.	[259]
	Decitabine (DNMTi) and Panobinostat (HDACi)	In combination with alkylating agent temozolomide showed great improvements in disease stabilization and remission.	Resistant metastatic melanoma patients	[260]
	Azacitidine and Valproic acid	In combination with carboplatin demonstrates decreased DR4 methylation and shows modest evidence of antitumor activity	Patients with heavily treated advanced ovarian cancer.	[261]
	BRD4i (BRD4 inhibitor)	Induced homologous recombination deficiency and sensitized cells to PARP inhibition.	Multiple tumor lineages regardless of BRCA1/2, TP53, RAS or BRAF mutation status in vitro and in vivo.	[262]

Inhibiting AKT has for long been a major focus as a promising therapeutic approach in cancer. To date, there are two classes of AKT inhibitors, namely, ATP-competitive and allosteric inhibitors which either block ATP binding or prevent AKT phosphorylation and activation [263]. However, despite the development of many compounds with promising results for targeting AKT, none of these inhibitors has been approved yet for clinical use. This is, at least in part, attributed to the complex and pleiotropic functions that AKT exerts in cells. Therefore, combinational therapy approaches seem to represent a major research direction for the successful clinical utilization of AKT inhibitors [264].

Conventional drugs, such as cisplatin and doxorubicin, exert their anti-cancer effects via the accumulation of ROS and DNA damage. However, reduction of ROS generation leads to resistance [265]. Moreover, synthetic agents tested against MDR cancers, can sometimes cause toxic side effects and lack specificity. For these reasons, efforts have also been focused on natural agents and their derivatives, to take advantage of their beneficial properties. Polyphenols represent a large family of organic, naturally occurring compounds that are characterized by the presence of many phenol groups in their structure. Phenolic compounds, including flavones, ellagitannins and curcumin, are known to act as chemopreventive agents due to their antioxidant properties and their ability to inactivate pro-carcinogens. Certain natural compounds, rely on the upregulation of ROS to induce DNA damage. However, low oxygen levels within the tumor leads to limited generation of ROS and allows cancer cells to escape death [266]. Reduced ROS levels in cancer cells have been reported to increase the levels of P-gp efflux pump through the JNK pathway, further promoting drug resistance [267]. Importantly, natural compounds also exert chemotherapeutic properties because they can regulate signaling pathways to inhibit the proliferation of cancer cells, block angiogenesis and metastasis, and induce immune and inflammatory responses [268]. Importantly, phenolic compounds can induce apoptosis in cancer cells by activating various pro-apoptotic machineries and, interestingly, several have been reported to be effective against MDR tumors [269]. The TME is also being targeted to improve drug efficacy in difficult to treat cancers; immune checkpoint inhibitors (ICIs) that have shown promising clinical efficacy, are now being tested in combination with other agents to overcome intrinsic or acquired tumor resistance [270].

## 6. Future Perspectives: The Implication and Therapeutic Exploitation of Epigenetics in MDR

For decades, genetic mutations during cancer progression and acquired MDR were considered a major cause of treatment failure in relapsed cases, ignoring the non-genetic basis of tumor heterogeneity [271]. A breakthrough in anti-cancer therapy was achieved when it was realized that the high predominance of MDR is attributed not only to DNA mutations but also to a variety of epigenetic alterations. Moreover, it became increasingly evident that the probability of an irreversible mutation to appear, increases as more critical modifications emerge in the epigenome of tumor cells [272]. Among the observations that led to this conclusion were the reversal of drug resistance observed upon drug-free periods, the frequent absence of mutations in drug targets or activated pathways, as well as the heterogeneity in acquired MDR and in relapsed cases [273,274]. Recent data highlight the major role of epigenetic changes in tumorigenesis and in the development of MDR [275]. Cancer cells can escape from a poised drug-tolerant condition and enter into an epigenetically fixed acquired-resistant state via poorly understood mechanisms. During carcinogenesis, environmental pressure upon tumor cells results in an array of epigenetic aberrations, such as DNA and RNA methylation, alterations in the miRNAs expression and histone modifications, which eventually lead to epigenetically-induced transcriptional adaptation [272,275].

Epigenetic changes are generally reversible and susceptible to external factors; these characteristics make them appealing targets either for monotherapy or in combination with other anti-cancer agents to treat MDR [276,277]. Therefore, multiple generations of drugs that target epigenetic regulators, called epi-drugs, have been designed during the last 40 years, demonstrating valuable effects on cancer therapy in clinical trials [278,279]. Epigenetic modifications such as DNA hypermethylation of gene promoters could partially explain the acquired resistance after prolonged treatment [280,281]. Recent studies have shown that epi-drugs, such as the DNA methylation inhibitor (iDNMT), 5-aza-20-deoxycytidine (decitabine; DAC), can be effective against resistant cancers (including lung cancer and AML), in combination with conventional chemotherapeutics by reversing DNA methylation which sensitizes cancer cells to other chemotherapeutics, including carboplatin, cisplatin and 5-FU [282,283,284]. 

Several challenges remain, however, to improve the effectiveness of epi-drugs against MDR cancers. In contrast to hematological malignancies, solid tumors do not respond well to epi-drugs possibly due to the contribution of the TME [285]. To overcome these obstacles, single-cell sequencing technologies (i.e., scRNA-seq, scATAC-seq, sc-Hi-C and scChIP-seq) using patients’ biopsies immediately before and/or after epi-drug administration could provide more detailed information about the drug resistance landscape derived from genome/epigenome interactions. In addition, many epi-drugs, such as HDACs inhibitors which were shown to be effective against hematological malignancies and MDR, result in numerous off-target effects, since they act as pan-HDAC inhibitors [286]. A novel therapeutic approach, called proteolysis-targeting chimera (PROTAC) has been developed to improve the specificity of drugs against targets at low concentrations. The PROTACs technology promotes proteasomal protein degradation through E3 ubiquitin ligase activity. Importantly, side effects are significantly reduced [57]. PROTAC drugs targeting the epigenetic reader BRD4 are under preclinical evaluation in various cancer types showing promising therapeutic effects in reversing drug resistance phenotypes [57]. In addition, the replacement of preclinical models, such as two-dimensional (2D) in vitro cancer cell lines, with more clinically relevant 3D in vitro or mouse models that recapitulate the TME of resistant cells in relapsed patients should be considered [287,288]. Importantly, epigenetic interpatient and intratumor heterogeneity, a hallmark of human cancers that plays crucial roles in developing MDR need to be more thoroughly investigated [271,277]. Taken together, current evidence suggests that to improve efficacy of epi-drugs, new generations of more selective agents should be developed with optimized drug dosage, pharmacodynamics and pharmacokinetics properties, along with low toxicity levels in normal cells and tissues.

## 7. Conclusions

Deregulation of the major apoptotic pathways, and the related survival pathways that control the expression and/or activation of apoptotic proteins, may lead to MDR. Imbalance in Bcl-2 family levels, overexpression of IAPs and p53 inactivation have been widely reported in various types of MDR tumors. The TME appears to be pivotal in tumor progression and is known to impair the effectiveness of many therapeutics. A better understanding of how epigenetic alterations control cancer development may lead improved drug efficacy and contribute to the improvement of several agents already employed against MDR tumors. Recent studies using in vitro models have shown that different MDR mechanisms can be derived from a single ancestor cell [289]. Subsequently, in these cases, detection and characterization of residual tumor cells using single-cell sequencing technologies, will elucidate the contribution of genetic and epigenetic variability in developing drug resistance phenotypes [290,291]. It has been suggested that one of the better strategies in the fight against acquired MDR could be a therapeutic scheme that targets cancer cells prior to the acquisition of drug resistance, i.e., before they express a pro-survival program [280]. At the dawn of personalized medicine, the study of these alterations could provide novel and promising predictive biomarkers with great clinical significance against the evolution of acquired MDR.

## Figures and Tables

**Figure 1 cancers-13-04363-f001:**
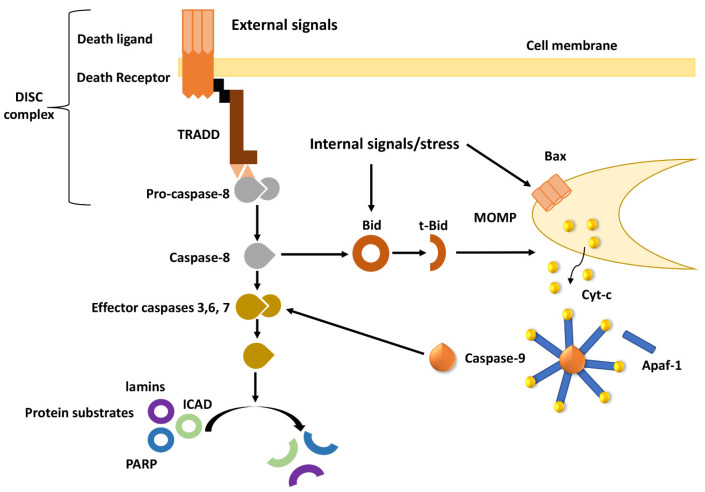
**The extrinsic and intrinsic pathways of apoptosis.** The activation of the extrinsic pathway involves binding of an external ligand to a transmembrane Death Receptor which then induces its conformation as a homotrimer. In the internal part of the receptor, the exposed Death Domain (DD) recruit adaptor proteins such as TRADD. Through its Death Effector Domain (DED), TRADD can then recruit pro-Caspase-8 which is activated through self-proteolysis. Subsequently the death ligand, receptor, TRADD molecule and caspase-8 form the DISC complex. Active caspase-8 can cleave downstream effector caspases 3, 6 and 7 which degrade nuclear lamins and other cellular components. Caspase-8 connects the extrinsic pathway with intrinsic or mitochondrial apoptotic signaling, though the cleavage of Bid to truncated Bid (t-Bid). Bid, a member of the Bcl-2 family, facilitates the opening of mitochondrial pores in a process called MOMP by inducing the polymerization of Bax on the outer mitochondrial membrane. MOMP is also induced by internal signals, such as extensive DNA damage. Cytochrome-c (cyt-c) is released from mitochondria and along with Apaf-1 and pro-caspase-9 form the apoptosome heptamer structure. Following proteolysis, caspase-9 cleaves and activates effector caspases, further amplifying the apoptotic process. Apoptotic Protease Activating Factor 1 (APAF-1); Mitochondrial Outer Membrane Permeabilization (MOMP); TNFR1-Associated Death Domain protein, (TRADD).

**Figure 2 cancers-13-04363-f002:**
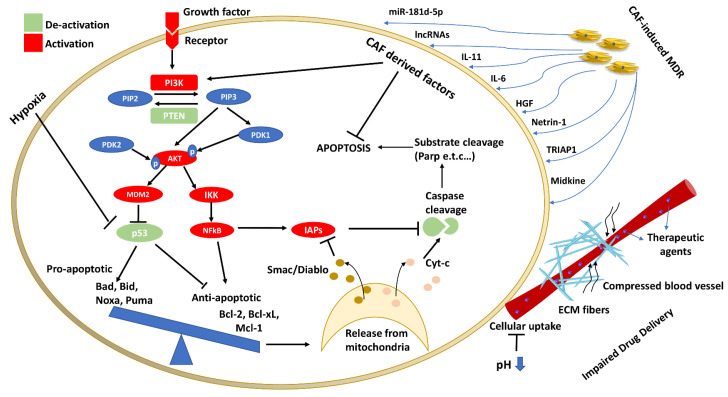
Deregulation of apoptotic pathways during the MDR development in cancer. The PI_3_K/AKT survival pathway is frequently overactive in MDR cancers, partly due to increased levels of growth ligands and receptors. Binding of growth factors to receptor tyrosine kinases (RTKs) stimulates PI_3_K by autophosphorylation; PI_3_K then catalyzes the conversion of PIP2 to PIP3 while tumor suppressor PTEN has an opposing function. PTEN is often mutated and thus inactive in MDR cancers. PI_3_K mediates the activation of AKT via phosphorylation on Thr^308^ and Ser^473^ by PDK1 and PDK2, respectively. AKT can then activate MDM2 which blocks the function of p53. P53 regulates (among others) the levels and activation status of the Bcl-2 family of proteins. The balance between the levels of the pro-apoptotic and anti-apoptotic Bcl-2 family proteins, controls the release of pro-apoptotic factors from mitochondria. Once released, cytochrome-c activates firstly caspase-9 and then the executioner caspases-3, -6 and -7. Furthermore, in MDR cancers, members of the IAP family are overexpressed, blocking caspase function. SMAC/Diablo are also released from the mitochondria during apoptosis and can inhibit the function of IAPs. NFkB is indirectly activated by growth factors via the PI_3_K/AKT pathway and up-regulates IAPs, e.g., survivin via regulation at the transcriptional level. Factors released by CAFs increase tumor cells survival via the activation of the PI_3_K/AKT pathway and inhibit apoptotic pathways. The cellular and non-cellular components of the TME as well as modifications in metabolic pathways and mechanical stress have been also implicated in resistance to cancer-cells targeting pro-apoptotic therapeutic agents. The composition and structure of stromal components in tumors increase interstitial fluid pressure (IFP) hindering the penetration of macromolecules through tissue and influence the sensitivity of tumor cells to therapy [141]. The hypoxic TME favours cells that have lost sensitivity to p53-mediated apoptosis and that are deficient in DNA mismatch repair leading to resistance to platinum-based chemotherapeutic agents [142]. Moreover, the low extracellular pH in tumors decreases the cellular uptake of weakly basic drugs such as doxorubicin, mitoxantrone, vincristine and vinblastine [143]. Multi Drug Resistant (MDR); Receptor Tyrosine Kinases (RTKs); Phosphatidylinositol 4,5-bisphosphate PtdIns(4,5)P2, (PIP2); Phosphatidylinositol (3,4,5)-triphosphate PtdIns(3,4,5)P3, (PIP3); Phosphoinositide-dependent kinase, (PDK); Tumor Microenvironment (TME); Inhibitors of Apoptosis Proteins, (IAPs); Cancer Associated Fibroblasts, (CAFs).

## Data Availability

Not applicable.

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
