# Peer review of "Apoptosis Deregulation and the Development of Cancer Multi-Drug Resistance"

_cancers, 2021, doi:10.3390/cancers13174363_

Round 1

Reviewer 1 Report

In this review by Neophytou et al, the authors describe the role of apoptosis dysregulation in contributing to multidrug resistance (MDR), often associated with cancer treatment. Overall, the review provides a detailed overview on this topic, comprehensively covering the major pathways involved in apoptosis driven MDR. Suggestions for improvement for the current version are as follows:

  1. While the authors are diligent in describing the three major intrinsic pathways, it is very hard to grasp these interconnected signaling proteins without a visual to accompany the text. It would be very helpful to have illustrations summarizing major pathway signaling / interactions / players for the intrinsic & extrinsic pathways respectively.
  2. It would be helpful for the flow of the review to have separate sections with subheadings : IL-11, IL-6, Sox2, etc for each individual description in Section 4.1.

Author Response

Reviewer 1

1.While the authors are diligent in describing the three major intrinsic pathways, it is very hard to grasp these interconnected signalling proteins without a visual to accompany the text. It would be very helpful to have illustrations summarizing major pathway signalling/interactions/players for the intrinsic & extrinsic pathways respectively.

We have added Figure 1 that describes the extrinsic and intrinsic pathways and their interaction in this revised manuscript.

2.It would be helpful for the flow of the review to have separate sections with subheadings: IL-11, IL-6, Sox2, etc for each individual description in Section 4.1.

We have separated Section 4.1 with subheadings, distinguishing CAF-derived EVs, Interleukins, Sox2 regulation and other proteins released by CAFs that affect the response of cancer cells to therapy.

Reviewer 2 Report

Cancers-1339929 described the deregulation of apoptotic pathway in multi-drug resistance (MDR) cancer. The authors also described the involvement of the tumor microenvironment in apoptosis and MDR, and presented a therapeutic approach to induce apoptosis in MDR cancer. This review paper is meaningful in that it provides a well-organized and integrated understanding of the deregulation of apoptosis in cancer MDR. Points to be considered in this manuscript are presented below.

  1. In the introduction section, the authors presented the generation of ROS as a factor promoting the development of MDR in cancer. I also find the description of the involvement of ROS in MDR and apoptosis intriguing and need to be described in more detail in this manuscript.
  2. Section 4.1 describes the involvement of CAF-EVs in resistance to apoptosis. I think the authors could add more references to describe the role of non-coding RNAs (including miRNAs) in EVs in apoptosis and MDR.
  3. Most of the paragraphs consist only of a list of references throughout the manuscript. Discussion on the importance of the content of each section or paragraph and direction for further research needs to be added.

Author Response

Reviewer 2

1.In the introduction section, the authors presented the generation of ROS as a factor promoting the development of MDR in cancer. I also find the description of the involvement of ROS in MDR and apoptosis intriguing and need to be described in more detail in this manuscript.

We have added information relating to the role of ROS in affecting apoptosis, MDR and chemoresistance in lines 225-227, 544-546, and 554-558. Specifically, we included references that discuss the regulation of p53 transcriptional activity by ROS (ref 61), the importance of ROS in chemotherapy (ref 233), the effect of limited ROS levels because of hypoxic conditions within the tumor (ref 234) as well as the increase of P-gp levels in the presence of ROS (ref 235). 

2.Section 4.1 describes the involvement of CAF-EVs in resistance to apoptosis. I think the authors could add more references to describe the role of non-coding RNAs (including miRNAs) in EVs in apoptosis and MDR.

We have included four more references discussing the role on non-coding RNAs in regulating apoptosis and MDR (refs 162, 164-166).

3.Most of the paragraphs consist only of a list of references throughout the manuscript. Discussion on the importance of the content of each section or paragraph and direction for further research needs to be added.

We added explanatory sentences highlighting the importance of specific paragraphs, including future directions and sources for further reading in lines 268-271, 321-326, 418-421, 451-454 and 466-469.

Round 2

Reviewer 2 Report

The authors made improvements to the manuscript for all comments.